# Comparative Energetics of Various Membrane Distillation Configurations and Guidelines for Design and Operation

**DOI:** 10.3390/membranes13030273

**Published:** 2023-02-24

**Authors:** Md Rashedul Islam, Bosong Lin, Yue Yu, Chau-Chyun Chen, Mahdi Malmali

**Affiliations:** Department of Chemical Engineering, Texas Tech University, Lubbock, TX 79409-3121, USA

**Keywords:** membrane distillation (MD), gained output ratio (GOR), energetics, modeling, e-NRTL

## Abstract

This paper presents a comparative performance study of single-stage desalination processes with major configurations of membrane distillation (MD) modules. MD modules covered in this study are (a) direct contact MD (DCMD), (b) vacuum MD (VMD), (c) sweeping gas MD (SGMD), and (d) air gap MD (AGMD). MD-based desalination processes are simulated with rigorous theoretical MD models supported by molecular thermodynamic property models for the accurate calculation of performance metrics. The performance metrics considered in MD systems are permeate flux and energy efficiency, i.e., gained output ratio (GOR). A general criterion is established to determine the critical length of these four MDs (at fixed width) for the feasible operation of desalination in a wide range of feed salinities. The length of DCMD and VMD is restricted by the feed salinity and permeate flux, respectively, while relatively large AGMD and SGMD are allowed. The sensitivity of GOR flux with respect to permeate conditions is investigated for different MD configurations. AGMD outperforms other configurations in terms of energy efficiency, while VMD reveals the highest permeate production. With larger MD modules, utilization of thermal energy supplied by the hot feed for evaporation is in the order of VMD > AGMD > SGMD > DCMD. Simulation results highlight that energy efficiency of the overall desalination process relies on the efficient recovery of spent for evaporation, suggesting potential improvement in energy efficiency for VMD-based desalination.

## 1. Introduction

Water scarcity is a pressing issue that poses an urgent challenge to the world. Meanwhile, water quality problems are also increasing water-related health risks [1]. Approximately 80% of wastewater returns to the environment without being treated. As a result, roughly 3 million people die annually because of waterborne diseases and pollution [2]. Global water problems can be resolved through different water purification technologies, such as desalination [3] and adsorption [4], depending on water characteristics. Desalination of saline water streams, such as seawater, brackish water, and industrial wastewater, has played an important role in meeting increasing water demands [2]. Among several desalination technologies, membrane distillation (MD) has an advantage over others in that it can deal with high-salinity brine streams while using low-grade or waste heat [5,6]. MD is a thermally driven membrane separation process where a warm saline feed stream comes in direct contact with a hydrophobic membrane while a cool permeate stream is on the other side. Separation is achieved by evaporation of volatile water at the feed–membrane interface and preferential diffusion of water vapor through the membrane pores, which leads to 100% theoretical rejection of nonvolatile solutes [5,7,8,9]. The feature of evaporation below the boiling point extends the application of the MD process to treat hypersaline solutions with proper utilization of low-grade waste heat and augments the energy efficiency with the integration of renewable energy, e.g., geothermal heat and solar power [10,11,12,13]. Compared to commercial desalination technologies, such as reverse osmosis and multistage flash, however, MD inherently consumes more energy because of the difficulty in latent heat recovery, leading to low energy efficiency [6,14].

There have been various reports that systematically investigated the membrane flux and energy efficiency of different MD configurations [15,16,17,18,19]. Eykens et al. experimentally identified the suitable configuration and membrane selection under different working conditions, including feed temperature, flow rate, and salinity [19]. There are four commonly used MD configurations, depending on water vapor recovery approach on the permeate side: direct contact MD (DCMD), air gap MD (AGMD), sweeping gas MD (SGMD), and vacuum MD (VMD) [20]. Although the general order of fluxes has been confirmed as VMD > DCMD > SGMD > AGMD, each configuration was found to have different sensitivity to the specific system parameters and hence different working performance [19]. The energy efficiency of an evaporation-based desalination process is typically quantified as gained output ratio (GOR), defined as the ratio of the latent heat of evaporation over the amount of heat supplied to the system to produce 1 kg of water [6]. GOR is directly affected by filtration system parameters, such as membrane characteristics, module size (i.e., membrane area defined as width × length) and configuration, heat integration scheme, and flow conditions [15,16,17,18,19]. An ideal membrane for MD should have low resistance to vapor transfer and high resistance to heat transfer by conduction. Better thermal efficiency can be achieved with novel membrane design and optimization of process design and operations [6,21].

Rigorous modeling of the MD configurations aids design, development, and optimization of MD desalination processes. Summers et al. established detailed analytical models for different MD configurations based on heat and mass transfer through the membrane and conservation equations in the flow channels [22]. The models were validated with experimental data and used to examine the effect of module geometry and operating conditions on GOR in each single-stage MD system for seawater (35 g/kg) desalination. The results reveal that a higher GOR can be achieved in AGMD and DCMD in comparison to VMD. More attention was also paid to the use of MD in desalination of high-salinity water, as high water recovery and low volume of concentrated brine discharge are targeted by zero liquid discharge strategies [23]. Swaminathan et al. investigated energy efficiency (GOR) of different gap MD configurations (e.g., AGMD, permeate gap MD, and conductive gap MD) in a wide range of feed salinity up to 260 g/kg, as well as the sensitivity with respect to membrane thickness and system size [24]. An expression for GOR was derived as a function of the transmembrane temperature difference and vapor pressure depression, from which an optimal GOR value was found at the critical module size (length at fixed width) [24]. The result suggests that gap MDs perform better than DCMD in terms of energy efficiency and flux because of lower heat transfer resistance of the gap than that of the external heat exchanger in DCMD. The proposed thermal resistance network model provides a framework for comparative analysis of different MD configurations. Since heat and mass transfer occur simultaneously, it is interesting to know whether a critical module size can be determined based on the distinct mass transfer mechanism in different MD configurations, which can provide a different perspective on the design and operation of MD processes. Hence, the motivation of this study was to derive a general expression for mass transfer as a function of the temperature difference and system permeability to find the critical module size for feasible flux production.

Accounting for the nonideal behavior of the aqueous electrolyte solution in MD modeling research is important for improving the accuracy and fidelity of the simulation results. Currently, some desalination studies treat saline solutions as ideal solutions [25], which simply assume an ideal behavior for a nonideal electrolyte solution and do not consider the dissociation of the solute. However, the ionic components in the saline feed have a notable influence on the phase equilibrium properties, such as vapor pressure and solubilities of gases and salts, especially for high-salinity brines [16,26,27,28,29]. Different approaches have been proposed to take into account the nonideal behavior of the aqueous electrolyte solution, such as correlations for seawater thermodynamic properties [30], the Pitzer model [31], and the electrolyte NRTL models [32]. In comparison to other methods, the electrolyte NRTL model incorporates local composition theories and uses only two binary interaction parameters to describe the physical interactions for ions and molecules. More importantly, the model has proven to be reliable in predicting thermodynamic properties over a wide dissolved solid concentration range [26,33]. The electrolyte NRTL model is considered in this work to calculate the thermophysical and transport properties of hypersaline water due to its high accuracy and readily available model parameters [16,26,27,34].

In this study, the governing equations of DCMD, VMD, SGMD, and AGMD are formulated rigorously in Aspen Custom Modeler (ACM) V10.0 (Aspen Technology Inc, Bedford, MA 01730, USA), together with the electrolyte NRTL model to provide thermophysical and transport properties. The MD models are validated and exported to Aspen Plus V10.0 for the system-scale process modeling investigation. A comparative analysis of different MD configurations is presented through a general criterion, defined as a function of saturation temperatures. The GOR–flux performance curve of MDs is obtained and the sensitivity of performance metrics with respect to operating conditions is evaluated for each MD configuration. The implications are provided as intrinsic guidelines for the design and operation of the MD desalination process.

## 2. Material and Methods

### 2.1. Modeling of MD Modules

For the determination of comparative performance and energy efficiency of the different MD modules, detailed modeling of mass and energy transport involved in each configuration is critical. One-dimensional, steady-state models for various MD configurations with flat sheet geometry are developed from the mass, momentum, and energy balances in the MD channels, displayed in Figure 1. Countercurrent flow orientation is considered for DCMD, AGMD, and SGMD.

#### 2.1.1. Mass Transfer Mechanism in MD Configurations

This section briefly summarizes the details of mass transfer involved in MD processes [5,7,8]. The summary of mass transfer mechanisms involved in the four selected MD configurations is summarized in Table 1, which is discussed in the following.

The mode of mass transport through the membrane is primarily controlled by diffusion: Knudsen and molecular diffusion. Diffusion of vapor molecules experiences resistance from the collision with pore wall and stagnant molecules, i.e., air trapped inside pores. The vapor molecule–pore wall collision is analogous to free molecular diffusion. In free molecular diffusion, molecules of low-density vapor travel through the pore without colliding with other molecules but the pore wall; hence, the free molecular diffusion is described by Knudsen diffusion. In the molecular diffusion process, vapor molecules collide with air molecules; hence, the binary diffusion of vapor–air is best described using the molecular diffusion process. The Knudsen and molecular diffusion are expressed in Equations (1) and (2), respectively.
(1)$$J_{w}^{K}=\frac{1}{3}\frac{\varepsilon }{\chi }\frac{d_{p}}{\delta_{m}}\sqrt{\frac{8M_{w}}{\pi RT}}\Delta p_{w}$$
(2)$$J_{w}^{M}=\frac{D_{wa}M_{w}}{RT}\frac{\varepsilon }{\chi }\frac{\bar{P}}{{\left|p_{a}\right|}_{\ln}}\frac{\Delta p_{w}}{\delta_{m}},\;\mathrm{where}\;{\left|p_{a}\right|}_{\ln}=\frac{p_{a,f,m}-p_{a,p,m}}{\ln\frac{p_{a,f,m}}{p_{a,p,m}}}$$

Here $\Delta p_{w}$ is the partial pressure difference of water across the membrane at the membrane surface temperature on the feed and permeate sides. The definition of symbols is provided in the nomenclature. For a nonideal solution, partial pressure of water ($p_{w,f,m}$) at the feed–membrane interface is a function of water activity and saturated vapor, as reported in Equation (3).
(3)$$p_{w,f,m}=x_{w}\gamma_{w}p_{w}^{*}$$

In general, the Antoine equation is used to estimate the vapor pressure of water at saturation, $p_{w}^{*}$ [35]. Selecting the right thermodynamic model is critical to calculate the activity coefficient of water, $\gamma_{w}$, in which the nature of the solution (e.g., electrolytes) must be considered. Determination of partial pressure of water in the permeate side depends on the MD configuration, which is discussed in the following sections.

Apart from $\Delta p_{w}$, the properties of the membrane material and pore size largely influence the vapor flux. The ratio of the mean free path of water molecules ($\lambda_{w}$) and the pore size ($d_{p}$), typically defined as Knudsen number ($K_{n}$), determines the modes of diffusion. Equation (4) presents a stepwise function to summarize the effective diffusion flux for the entire regime with $K_{n}$ [36,37].
(4)$$J_{w}^{D}=\{\begin{array}{cc} J_{w}^{K} & ,\;K_{n}>1\\ {\left[\frac{1}{J_{w}^{K}}+\frac{1}{J_{w}^{M}}\right]}^{-1} & \;\;\;\;\;\;\;\;\;\;\;\;,\;0.01\leq K_{n}\leq 1\\ J_{w}^{M} & \;\;\;\;,\;K_{n}<0.01 \end{array}$$

For the case of VMD (Table 1), there exists a total pressure gradient across the membrane causing an advective vapor flow, which enhances the total flux. The advection flow is analogous to the viscous flow through a cylindrical (capillary) tube. As reported in Equation (5), the viscous flow can be modeled as Poiseuille flow. In VMD configuration, the removal of air from the membrane porous structure obviates the molecular diffusion. Therefore, VMD involves Knudsen diffusion and viscous flow. A stepwise function of $K_{n}$ is used to define the mass transport mechanism regimes in VMD. For a normal operating range of vacuum pressure, i.e., 3–50 kPa, in the VMD process, $K_{n}$ can be an order of magnitude larger than other configurations [38].
(5)$$J_{w\;}^{V}=\frac{1}{32}\frac{\varepsilon }{\chi }\frac{d_{p}^{2}}{\delta }\frac{M_{w}{\bar{P}}_{\;}}{\mu_{v}RT}\Delta p_{w}$$
(6)$$J_{w}^{T}=\{\begin{array}{cc} J_{w}^{K},\;\;\;\;\;\;\;\; & Kn>10\;\;\;\;\;\;\;\;\;\;\;\;\;\;\\ J_{w}^{K}+J_{w}^{V}, & 0.01\leq Kn\leq 10\\ J_{w}^{V},\;\;\;\;\;\;\;\;\;\; & Kn<0.01\;\;\;\;\;\;\;\;\;\;\; \end{array}$$

In the following, we first elaborate on details of transport on the feed side for all the configurations as feed side equivalent for all the selected configurations. We then discuss the details of the model for the permeate side in each configuration.

#### 2.1.2. Transport in the Feed Channel

The common feature in all four MD configurations is the feed channel. The hot feed stream supplies the heat of vaporization and accompanying heat loss through the membrane, and the heat transfer by convection is due to the temperature gradient across the boundary layer adjacent to the membrane surface. The hydrophobic membrane theoretically offers 100% rejection for dissolved solids. Therefore, electrolyte concentration builds up on the surface of the membrane, which leads to a concentration gradient across the boundary layer, also known as concentration polarization. The transport in the feed channel is described using a set of mass, momentum, and energy balance equations (Equations (7)–(10)) over the control volume, as depicted in Figure 1.
(7)$$\frac{dv_{f}}{dz}=-\frac{v_{w}}{h}$$
(8)$$\frac{d\left(v_{f}C_{f,b}\right)}{dz}=0$$
(9)$$\frac{dP_{f}}{dz}=-\frac{1}{2}f_{f}\frac{\rho_{f}}{d_{h}}v_{w}^{2}$$
(10)$$\rho_{f}C_{p,f}\frac{d\left(v_{f}T_{f,b}\right)}{dz}+\frac{q_{f}}{h}=0$$
Here, $v_{w}$ is the water vapor velocity perpendicular to the membrane ($v_{w}=J_{w}/\rho_{w}$) and $f_{f}$ is the dimensionless friction factor in the feed channel. The set of differential equations is solved for the inlet feed conditions. Similar set of balance equations is developed for the permeate channel of DCMD and SGMD, and coolant channel in AGMD, as presented in Table 2. Note that $J_{w}$ represents mass flux through the membrane in general ($J_{w}^{T}$ in VMD and $J_{w}^{D}$ in other MDs).

The heat transfer through the boundary layer ($q_{f}$) determines the temperature on the surface of the membrane. The heat transfer coefficient in the boundary layer is computed from the definition of Nusselt number (Nu). This modeling effort follows an empirical correlation of Nu with Re and Pr [39].
(11)$$q_{f}=h_{f}\left(T_{f,b}-T_{f,m}\right)$$

#### 2.1.3. Transport in the Permeate Channel

##### DCMD

The permeate side of the DCMD module is very similar to that of the feed side. Pure water flowing through the permeate channel provides a condensing surface for the permeating vapor. The distillate stream receives heat from the condensing vapor, as well as the heat conducted through the membrane, and the heat transport is defined as:(12)$$q_{p}=J_{w}\Delta H_{V,w}+\frac{k_{m}}{\delta_{m}}\left(T_{f,m}-T_{p,m}\right)$$

The condensing vapor is in equilibrium with pure water at the membrane surface temperature on the permeate side ($T_{p,m}$). Thus, the partial pressure of water at the permeate side is the saturated vapor pressure of water at $T_{p,m}$.
(13)$$p_{w,p,m}=p_{w}^{*}\left(T_{p,m}\right)$$

Variation of process variables in the permeate channel is estimated from the mass, momentum, and energy balance over a control volume, depicted in Figure 1. The balance equations are summarized in Table 2.

##### VMD

The MD flux is enhanced by creating vacuum in the MD permeate chamber. The vapor extraction from permeate chamber is achieved using a vacuum pump. During startup, the vacuum pump removes the trapped air from the membrane and chamber. At steady-state operation, membrane pores and permeate chamber are completely occupied by water vapor, following the conservation of mass in the system. Therefore, partial pressure of water at the membrane–permeate interface is the vacuum pressure ($P_{p}$). The modes of mass transfer involved in the VMD process are described in a previous section. The overall flux in VMD depends on the membrane surface temperature on the feed side ($T_{f,m}$) and the vacuum pressure. At reduced pressure, the water vapor remains in the vapor phase.
(14)$$p_{w,p,m}=P_{p}$$

The heat flux from the feed side boundary layer ($q_{f}$) is mostly used to vaporize water. Water vapor undergoes an isothermal expansion in vacuum as it passes through the pore [22]. Thus, the vapor temperature at membrane–permeate interface is $T_{f,m}$. With the permeate under vacuum, conductive heat loss through the membrane is insignificant. However, a portion of $q_{f}$ is used for the isothermal expansion of vapor ($q_{exp}$), which is assumed to be negligible compared to the heat flux for vaporization, i.e., $J_{w}\Delta H_{V,w}\gg q_{exp}$. Overall permeate production can be calculated from the integration of fluxes of control volumes along the feed channel.
(15)$${\dot{m}}_{p}=\int_{0}^{L}J_{w}dzW$$

##### SGMD

In the SGMD process, a sweeping gas (i.e., air) flows through the permeate channel to reduce the water partial pressure. The partial pressure of water on the permeate–membrane interface ($p_{w,p,m}$) depends on the mole fraction of water ($y_{w}$) in the air–vapor mixture in the permeate channel. The mole fraction of water vapor is a function of moisture content, mass flow rate of dry air, and the mass flux of water vapor through the membrane [20].
(16)$$p_{w,p,m}=y_{w}P$$
(17)$$y_{w}=\frac{\omega }{\omega +0.622}$$
(18)$$\omega =\omega_{in}+\frac{J_{w}dA}{{\dot{m}}_{a}}$$
where ω and $\omega_{in}$ are the instantaneous and initial humidity ratios, respectively, and ${\dot{m}}_{a}$ and dA are the mass flow rate of dry air and active area of the membrane in the control volume, respectively.

The water evaporated at the feed–membrane interface enters the permeate channel and remains in the vapor phase. Therefore, the heat conducted through the membrane will exchange sensible heat with moist air. The heat flux through the boundary layer in the permeate channel is equivalent to the heat of conduction through the membrane.
(19)$$q_{p}=h_{p}\left(T_{m,p}-T_{b,p}\right)=\frac{k_{m}}{\delta_{m}}\left(T_{f,m}-T_{p,m}\right)$$

##### AGMD

The exclusive feature of AGMD is that an air-filled chamber, generally known as an air gap, is in contact with a coolant channel. A condensing plate separates the air gap and coolant channel. Water vapor leaving the membrane eventually condenses on the plate. A thin film of condensate is then formed on the plate. Therefore, water vapor experiences an additional molecular diffusion in the air gap. The overall vapor flux is the result of diffusion flux through the membrane and molecular diffusion flux in the air gap [15,40]. Thus, the water vapor partial pressure difference at the feed–membrane interface and the condensing plate is the driving force for the overall flux. The partial pressure of vapor at the condensing plate is the saturated vapor pressure of water at $T_{g,fl}$.
(20)$$J_{w\;}^{\;}={\left[\frac{\delta_{m}}{D_{w}^{m}}+\frac{y_{a}\left(\delta_{g}-\delta_{fl}\right)}{D_{wa}}\right]}^{-1}\frac{\left(p_{w,f,m}-p_{w,g,fl}\right)}{RT}\;M_{w}$$
Here, $D_{w}^{m}$ is the effective diffusion coefficient of vapor in the membrane, which can be estimated from Equation (4), and $y_{a}$ is the average mole fraction of air in the membrane and air gap. Film thickness ($\delta_{fl}$) is calculated from the force balance around a control volume of the falling film [41]. However, the film thickness is very much smaller than the air gap thickness, i.e., $\delta_{g}\gg \delta_{fl}$.
(21)$$\delta_{fl}={\left[\frac{3\mu_{fl}\int_{0}^{z}J_{w}\left(z\right)dz}{\rho_{fl}\left(\rho_{fl}-\rho_{wa}\right)g}\right]}^{\frac{1}{3}}\;$$

The thickness of the air gap ($\delta_{g}$) in the control volume is very small compared to membrane length and width. Convective flow due to natural circulation along the length and width is negligible. The heat received by the conduction through the membrane will be conducted to the condensate film, and the condensate film recovers the heat of vaporization. Therefore, heat flux to condensate film is $q_{f}$. However, a portion of heat is dissipated with the condensate ($\dot{m_{p}}$). The condensate heat content is calculated at the average condensate temperature ($T_{cond}$) from an arbitrary temperature 0 °C. Due to the small film thickness, heat transfer through the film is via conduction. The heat flux ($q_{c}$) through the condensing plate is calculated following Equation (22). The thermal conductivity of the plate determines the plate’s coolant side temperature. The temperature difference across the plate is negligible, and $q_{c}$ is transferred through the boundary layer adjacent to the plate.
(22)$$q_{c}=q_{f}-J_{w}C_{p,c}T_{cond}$$

### 2.2. Critical Module Size Determination

This section establishes a general criterion to find a critical module size (*L* × *W*) for feasible flux production in terms of saturation temperature. As reported in Equation (23) the overall vapor flux in MD processes can generally be expressed as the product of permeability (B) and the partial pressure difference ($\Delta p_{w}$) across the vapor transport region. The permeability (B) is basically membrane permeability ($B^{m}$), except for AGMD. Membrane permeability results from the permeabilities due to Knudsen diffusion and molecular diffusion of vapor in the vapor–air mixture. Vapor experiences an additional mass transfer resistance in the air gap of AGMD due to vapor–air binary diffusion. The vapor permeabilities in MD processes are summarized in Table 3. B is a strong function of membrane characteristics and thickness of the air gap. For typical operating conditions of MD, B is a weak function of diffusion coefficients and mole fraction of air trapped in the membrane and air gap. For a specific membrane and air gap, a constant value for B can be assumed considering an average temperature ($\bar{T}$) in the vapor transport region.
(23)$$J_{w}^{\;}=B^{\;}\Delta p_{w}$$

The feed-side temperatures decrease along the length of the membrane due to water evaporation at the feed–membrane interface. As a result, the driving force ($\Delta p_{w}$) keeps decreasing with the temperature drop. For pure water in the feed of DCMD, $\Delta p_{w}$ approaches zero for the infinitely large membrane as the temperature on the feed side of the membrane ($T_{f,m})$ approaches the temperature on the permeate side ($T_{p,m})$. However, in the practical application of MD in desalination, $p_{w,f,m}$ is further reduced due to increased feed salinity. For a particular feed salinity, there is a critical length at which $p_{w,f,m}=p_{w,p,m}$, and beyond the critical length $p_{w,f,m}<p_{w,p,m}$ leading to a reverse flux. As depicted in Figure 2, the vapor pressure of saline water at $T_{f,m}$ is equivalent to saturated vapor pressure of water while $T_{w,f,m}^{*}$ is the effective saturation temperature at the feed–membrane interface. The vapor pressure depression results in a saturation temperature drop of $\Delta T_{w}^{\;}$ in the feed side. The permeate side water vapor pressure ($p_{w,p,m}$) in DCMD is saturated at $T_{p,m}$. Therefore, the partial pressure difference in Equation (23) can be replaced by a function of saturation temperature by approximating the exponential nature of vapor pressure as $p^{*}=ae^{bT}$.
(24)$$J_{w}=Bae^{bT_{p}^{*}}\left\{e^{b\left(\Delta T_{m}^{\;}-\Delta T_{w}^{\;}\right)}-1\right\}$$
Here, $T_{p}^{*}$ is the equivalent saturation temperature for partial pressure of water in the permeate side. $\Delta T_{m}^{\;}$ is the difference between $T_{f,m}$ and $T_{p}^{*}$, while $\Delta T_{w}^{\;}$ is the difference between $T_{f,m}$ and $T_{w,f,m}^{*}$. *a* and *b* are the parameters for the exponential function of water vapor pressure (adapted from the exponential curve fitting of vapor pressure vs. temperature from the Antoine equation in Figure 2 [35]). Forward flux will continue as long as $\Delta T_{m}^{\;}\geq \Delta T_{w}^{\;}$. Equation (24) is generally applicable for all four configurations of MD. As illustrated in Figure 2, $T_{p}^{*}$ is equivalent saturation temperature for the partial pressure of vapor ($p_{w,p,m}$) in air–vapor mixture in SGMD and vacuum pressure ($P_{p}$) in VMD. Since vapor is condensing on the plate in AGMD, $T_{p}^{*}$ is equivalent to $T_{g,fl}$. $T_{p}^{*}$’s for MD modules is summarized in Table 4. $\Delta T_{m}^{\;}$ decreases along the length of the membrane, and therefore the critical module size can be identified when $\Delta T_{m}^{\;}\left(z\right)\approx \Delta T_{w}^{\;}$.

### 2.3. Implementation of MD Models

*Model development*. The modeling procedure for the performance evaluation of MD-based desalination processes is outlined in Figure 3. Significant effort was directed towards the development of a one-dimensional steady-state model for DCMD, VMD, SGMD, and AGMD from mass, momentum, and energy balance equations and appropriate modes of mass and heat transfer involved. Countercurrent flow configuration was considered for DCMD, SGMD, and AGMD. All of these MD modules are developed in ACM [42]. Aqueous NaCl solution was considered as the feed to simulate the effect of feed salinity. An e-NRTL model was used for aqueous NaCl solution to support the calculation of thermophysical and transport properties [26]. Model parameters for the aqueous NaCl system were retrieved from literature [16,27]. MD modules were exported to the Aspen Plus library for the development of desalination processes with various MD modules and subsequent evaluation of their performances [43]. Flowcharts of desalination processes in Aspen Plus integrate the e-NRTL thermo-package to calculate thermophysical and transport properties during simulation.

*Model validation*. After the development of MD modules, each MD module was validated against the experimental results collected from the literature [15,16,17,18]. Module geometry (length × width × height), membrane type and characteristics, operating temperature and pressures, and feed salinity are compiled in Table 5. With specified experimental conditions, the MD process was simulated for the calculation of permeate fluxes. The calculated fluxes were compared with the available experimental results in terms of percentage average relative deviation (%ARD), following Equation (25). As indicated by %ARD in Table 5 (see the calculated and measured water fluxes for each MDs in Appendix A and [18]), the calculated flux is in good accordance with the measured flux for each MD module.
(25)$$\%ARD=\frac{1}{n}\overset{n}{\underset{i=1}{\sum }}\frac{\left|{\hat{J}}_{w,\;i}-J_{w,\;i}\right|}{{\hat{J}}_{w,i}}\times 100$$
Here, $\hat{J}$ and J are measured and calculated water fluxes, respectively, for a particular MD, i is the data point index, and n is the total number of data points.

## 3. Results

### 3.1. Critical Module Size of MD Configurations

As summarized in Table 6, each MD module was simulated for the feed flow rate of 1634 kg/h and temperature of 85 °C. The channel width was set as 6 m. Airflow with 30% relative humidity (RH) was considered in the permeate channel of SGMD. Vacuum pressure of 3.5 kPa was applied in the permeate chamber of VMD. Figure 4 shows the variation in $\Delta T_{m}$ and $\Delta T_{w}$ along the channel length for the feed salinity ranging from 60 to 300 g/L of NaCl. In all the MD configurations, $\Delta T_{m}$ and $\Delta T_{w}$ decrease along the channel length as separation continues. A maximum $\Delta T_{m}$ at the entrance indicates a large driving force, which is due to the enhanced concentration polarization at the entrance. For DCMD, $\Delta T_{m}$ and $\Delta T_{w}$ curves intersect, leading to a critical length at different feed salinities. Reverse flux will occur for a system longer than the critical length. The critical length (width fixed at 6 m) decreases from ~7 to ~2 m as the feed salinity increases from 60 to 300 g/L. A similar finding is observed for VMD. A relatively high mass flux results in a great heat transfer rate through the membrane surface, leading to a rapid drop in $\Delta T_{m}$ towards $\Delta T_{w}$. Because of the dominant effect of the high flux on $\Delta T_{m}$, the critical length is independent of the feed salinity. In SGMD and AGMD, a similar trend can be seen on $\Delta T_{m}$ and $\Delta T_{w}$; however, the curves do not intersect within the channel length of interest, which suggests that the driving force can be maintained for a large module.

### 3.2. Sensitivity with Permeate Conditions

With critical module size evaluated for different MD configurations in the previous section, performance metrics are evaluated with respect to permeate conditions. Thermal performance is quantified in terms of GOR. It is worth noting that GOR measures the thermal performance of the entire process rather than the MD modules themselves. A base case desalination process is developed with each configuration of MD, and the flowchart of the base case DCMD-based desalination process is presented in Figure 5. The feed stream (S1) is heat integrated with the permeate stream (S6) via a shell-and-tube heat exchanger (B1). Stream S2 exiting B1 is further heated to reach the target feed temperature. Flowcharts with VMD, SGMD, and AGMD modules are provided in Appendix B. The geometry of each module and operating conditions for each desalination process are reported in Table 7. As outlined in Equation (26), GOR for the base case desalination process is calculated from the ratio of the heat required to vaporize the permeate produced to the total heat input to the process.
(26)$$GOR=\frac{{\dot{m}}_{p}\Delta H_{V,w}}{{\dot{Q}}_{in}}$$

Here ${\dot{m}}_{p}$ is the permeate production (the difference between mass flow rates of stream S6 and S4 in Figure 5), $\Delta H_{V,w}$ is the latent heat of vaporization of water, and ${\dot{Q}}_{in}$ is the external heat input in the heater (B2).

**Figure 5 membranes-13-00273-f005:**
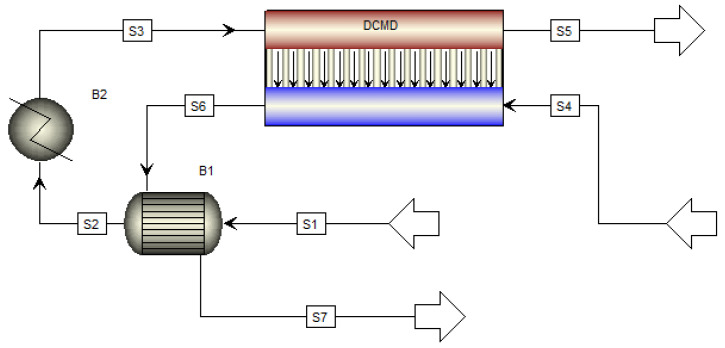
Flowchart of a base case DCMD-based desalination process.

**Table 7 membranes-13-00273-t007:** Geometry of the modules and operating of single-stage MD desalination processes.

MD Modules	DCMD	VMD	SGMD	AGMD
Module geometry				
Length (m)	4	1–3	12	12
Width (m)	6	6	6	6
Channel height (mm)	2	2	2	2
Air gap (mm)	–	–	–	1–4
Operating condition				
Feed (feed temperature, flow rate, salinity)	20 °C (S1), 85 °C (S3), 1634 kg/h, and 120 g/L			
Permeate	Distillate	Vacuum	0-90% RH	Coolant
	20–40 °C	3.5–8.5 kPa	20 °C	20–50 °C
	1200–2000 kg/h		400–1300 kg/h	1250–2000 kg/h

#### 3.2.1. DCMD

Distillate condition for DCMD was varied for the distillate flow rate of 1200–2000 kg/h and temperature of 20–40 °C. The feed salinity of 120 g/L and critical length of 4 m were used. The GOR–flux performance curve is displayed in Figure 6. The permeate flux increases with the increase in the distillate flow. This is because of the enhanced driving force ($\Delta p_{w}$) at the increased distillate flow (see Equation (24)). More importantly, an optimal value of GOR is seen at different distillate temperatures, which can be explained by the relative increment of mass flux and heat supply. As the distillate flow increases, the distillate outlet temperature decreases, which inevitably decreases the heat recovered in the heat exchanger and increases the heat duty in the heater. Since flux increases more rapidly than the heat duty, GOR appears to increase to an optimal value. In contrast, the change in the rate of heat input surpasses that of the flux after a certain distillate flow rate, thus leading to a decrease in GOR after surpassing the optimal value. The same GOR–flux pattern is observed at different distillate inlet temperatures, and a relatively large GOR is found at a lower distillate temperature. This is attributed to the enhanced driving force ($\Delta p_{w}$) at a lower distillate temperature.

#### 3.2.2. VMD

The response of GOR–flux performance with respect to vacuum pressure in VMD is evaluated and shown in Figure 7. Clearly, VMD produces an order of magnitude larger flux than DCMD. The flux production in VMD decreases as permeate pressure increases because of the reduced driving force. As suggested in Figure 4b, the module size also plays an important role in VMD; therefore, a lower flux is observed in a larger module size due to the reduced driving force ($\Delta p_{w}$). VMD has a very low GOR compared to DCMD. The maximum GOR of 1.12 is achieved at vacuum pressure of 3.5 kPa in a 3-meter VMD module. A significant portion of the permeate channel (~90%) remains in the vapor phase while it still exchanges heat with the feed stream. This is because vacuum pressure in the permeate channel poses the main barrier to the full extraction of latent heat of vaporization from the permeate stream. A higher permeate pressure produces a lower flux and GOR because of the enhanced thermodynamic limitation for condensation.

#### 3.2.3. SGMD

The response of the GOR–flux performance with respect to permeate variables, such as airflow rate, air temperature, and moisture content is reported in Figure 8. The GOR–flux curve manifests a similar pattern in SGMD as that in DCMD. This is due to the interplay of mass flux and heat supply at varying airflow rates. The water flux is enhanced as the higher airflow increases, which is due to the enhancement of the driving force ($\Delta p_{w}$) associated with the lower relative moisture content in the permeate channel. At the same time, an increase in the airflow rate leads to a reduction in air outlet temperature. The reduced outlet temperature enhances the thermodynamic limitation for heat exchange with the feed stream, and thereby increases the overall heat duty in the heater. In other words, the rate of heat input in the heater also increases monotonically with increasing the airflow rate. Therefore, the GOR–flux curve reveals a maximum in Figure 8a. A similar GOR–flux profile is observed for different air moisture contents. A higher moisture content brings the air close to saturated air faster. Therefore, flux is decreased at higher moisture content and so is GOR. The air stream leaving the heat exchanger is saturated with water, and water vapor in the air–vapor mixture cannot be fully condensed in the heat exchanger and a significant fraction of the latent heat of vaporization remains unrecovered.

Figure 8b depicts the variation in the GOR–flux curve with respect to air inlet temperature and airflow rate, with the moisture content being fixed at 30% RH. The increased air inlet temperature brings $T_{p,m}$ closer to $T_{f,m}$, leading to a lower flux. Simultaneously, a higher air temperature leads to a higher air outlet temperature that facilitates greater heat transfer to the feed stream. GOR drops as air inlet temperature increases due to the dominant effect of reduced permeate production over the increased air outlet temperature.

#### 3.2.4. AGMD

The sensitivity of GOR–flux performance with respect to coolant temperature and flow rate and the thickness of the air gap is investigated in Figure 9. A salinity of 120 g/L NaCl is maintained in the coolant–feed stream. The coolant flow rate is fixed at 1634 kg/hr. As reported in Figure 9a, flux decreases with increasing coolant temperature as a result of the reduced driving force ($\Delta p_{w}$) in AGMD. Meanwhile, the elevated coolant inlet temperature increases the outlet temperature, which eventually reduces external heat input to the system. However, the reduction in heat input surpasses the reduction in permeate production; therefore, GOR increases with increasing coolant inlet temperature. A shorter air gap offers smaller mass transfer resistance. Therefore, vapor flux is enhanced at a reduced air gap. At a specific coolant inlet temperature, a minimum air gap of 1 mm in this study results in maximum GOR. A reduction in flux due to a thicker air gap eventually reduces the overall heat transfer feed to the coolant. Therefore, GOR decreases at a thicker air gap due to reduced mass and heat transfer. For every gap thickness, GOR tends to approach a maximum value.

The effect of the flow rate in the coolant–feed channel is addressed in Figure 9b. AGMD simulation is performed for a 1 mm air gap. The same flow rate passes through the coolant and feed channels, generating similar hydrodynamics in both channels. A high flow rate enhances overall mass and heat transfer which results in increased flux production. However, an increased flow rate increases the external heat load in the heater to a larger extent. Therefore, a reduction in GOR is observed at an increased flow rate. There exists a critical coolant temperature at which GOR reaches the maximum. 

### 3.3. Critical Gap Thickness in AGMD

Vapor transport in AGMD experiences resistance from Knudsen and molecular diffusion inside the porous membrane and molecular diffusion in the air gap. All mass transport resistance is effective in series. Since permeability is reciprocal of mass transport resistance, the vapor transport region with the lowest vapor permeability is the flux-controlling region. The permeabilities of vapor transport regions at varying gap thickness are plotted in Figure 10. Among all the permeability terms, vapor permeability in the air gap (*C*_gap_) appears the lowest, and it strongly depends on the gap thickness, showing a decreasing trend as air gap thickness increases. Vapor permeability in the membrane (*C*_mem_, calculated from *C*_KN_ and *C*_Mol_) is much higher than that of the air gap (*C*_gap_), regardless of the air gap thickness. Therefore, overall vapor permeability (*C*_ov_) in the AGMD process closely follows the permeability in the air gap (*C*_gap_). 

Vapor permeability in the air gap, as evident from Figure 10, controls the vapor flux. The vapor flux is then the product of the vapor pressure difference across the air gap and vapor permeability in the air gap. The profile of vapor pressure difference across the air gap is plotted together with vapor permeability in Figure 11. As the gap becomes larger, the vapor pressure difference ($\Delta p_{w}=p_{w,m,g}-p_{w,g,fl}$) across the air gap increases monotonically while vapor permeability decreases. Consequently, there exists a critical gap thickness for which the flux production is maximum. The film temperature strongly depends on the vapor flux since the vapor releases latent heat to the film upon condensation. Higher flux production by lowering the gap thickness is not possible because of the higher film temperature and the subsequent increase in vapor pressure ($p_{w,g,fl}$) in equilibrium with condensing liquid. Beyond the critical gap, the temperatures at feed–membrane and membrane–air gap interfaces decrease with higher flux production. Therefore, water vapor pressures, i.e., $p_{w,f,m}$ and $p_{w,m,g}$, decreases accordingly. Further reduction of gap thickness results in increased $p_{w,f,m}$ and $p_{w,m,g}$, and hence the vapor pressure differences (i.e., $\left(p_{w,f,m}-p_{w,g,fl}\right)$ and $\left(p_{w,m,g}-p_{w,g,fl}\right)$) become much smaller compared to their vapor permeability. This phenomenon can be observed in larger modules where the temperature in the feed channel can drop significantly. 

## 4. Discussion

The differences in permeate side configuration of MD modules bring differences in their mass and heat transport modes. Even with the same membrane in every module, it is difficult to make a direct comparison of performances among the MD configurations. Nonetheless, MD modules can be compared qualitatively in terms of individual thermal efficiency, the extent of heat recovery, and energy consumption in desalination processes per unit of water recovered.

In the MD module, part of the heat supplied by the hot feed through the boundary layer in the feed channel (${\dot{Q}}_{T}$) is conducted through the polymeric structure of the membrane. The conduction of heat across the membrane (${\dot{Q}}_{m}$) is considered heat loss. Thus, heat loss through the membrane is inevitable with an exception for VMD. Approximately, 20–80% of ${\dot{Q}}_{T}$ is likely lost through the membrane, depending upon the membrane properties, hydrodynamics of channels, and module size [44,45]. Additionally, a thinner memwith higher thermal conductivity contributes to increased heat loss at a low temperature difference across the membrane. This situation may lead to the reversal of flux when brine solution is treated in DCMD [46]. Thermal efficiency of individual MD modules significantly influences the thermal efficiency of the process. As outlined in Equation (27), the thermal efficiency (η) is defined as the ratio of the amount of heat used for the vaporization of water (${\dot{Q}}_{v}$) to the total heat supplied by the hot feed solution (${\dot{Q}}_{T}$).
(27)$$\eta =\frac{{\dot{Q}}_{V}}{{\dot{Q}}_{T}}$$

Thermal efficiencies of DCMD, SGMD, and AGMD are calculated at their respective operating conditions and illustrated in Figure 12. Operating conditions are the variation of (a) distillate flow in DCMD, (b) airflow in SGMD, and (c) coolant temperature in AGMD. Since the operation of MD configurations is different from each other, η is reported against flux. Other parameters follow the base condition as defined in Table 7. AGMD offers very high η (>0.8) since air gap reduces heat loss through the membrane. A higher coolant temperature elevates the temperature at the membrane–air gap interface and causes a reduction in conductive heat loss through the membrane. Similar η values for AGMD are reported in the literature [47]. High η (>0.8) is also observed for SGMD, which is comparable to that of AGMD. The boundary layer of air reduces the heat loss significantly. At a lower Re of air, the heat transfer coefficient of air boundary layer is reduced and consequently η increases. On the contrary, η for DCMD is much smaller, in the range of 0.22–0.44, which is in line with reported experimental results [45]. The higher heat transfer coefficient in the boundary layer of the permeate channel enhances heat transfer through the membrane. Permeate flux in DCMD increases with increasing the distillate flow rate. Overall, η increases with increasing the flux in DCMD: η of 0.94 is reported at the permeate flux of 22 kg/m^2^ h [47]. In the VMD process, isothermal expansion of vapor prevents heat loss through the membrane, and the heat of expansion is significantly smaller than the heat of vaporization. Therefore, η for VMD is considered 1. 

The heat integration scheme in the base case desalination process targets the maximum recovery of heat from the permeate stream to preheat the feed stream in an external heat exchanger. The vapor permeating through the membrane condenses within the membrane module in DCMD and AGMD. A permeate stream with high heat capacity in the DCMD process can preheat the feed stream entering the module. In AGMD, the heat recovery and integration take place within the module, thus obviating the use of an additional heat exchanger. In contrast, a small fraction of vapor is condensed from the permeate stream in VMD and SGMD. Additionally, the low heat capacity of vapor or air–vapor mixture contributes little to heat the feed stream. Figure 13 reports the fraction of vapor condensed in the external heat exchanger over the permeate flux produced. Variables here are the airflow for SGMD and module length for VMD. At permeate pressure of 3.5 kPa, only 9% of vapor is condensed. Thus, the fraction of partial condensation can be improved by raising the pressure in the heat exchanger. For SGMD, the fraction of vapor condensed varies from 5% to 20%. A higher airflow enhances the production of permeate flux and reduces the fraction of vapor in the air–vapor mixture. 

Figure 14 reports the specific thermal energy consumption (STEC) against permeate flux for each configuration. STEC is defined as the amount of thermal energy in kWh consumed in the desalination process to produce a unit m^3^ of distillate. Flux is used as a common variable among the MD configuration to observe the variations of their STEC. Variation of flux is generated by changing (a) distillate flow in DCMD, (b) permeate pressure in VMD, (c) airflow in SGMD, and (d) coolant temperature in AGMD. Desalination with AGMD offers the minimum STEC among all the MD modules as low as 79 kWh/m^3^. The high value of η of AGMD and efficient internal heat recovery results in such low STEC. At optimal operating conditions, DCMD can offer a minimum STEC of 190 kWh/m^3^. The efficient recovery of heat from the permeate stream in an external heat exchanger overcomes the poor thermal efficiency of the DCMD module. The opposite is observed in SGMD and VMD. The SGMD module, with much higher thermal efficiency, consumes more energy than DCMD to produce desalinated water. Poor extraction of heat from the vapor–air mixture due to partial condensation of water vapor raises the eventual energy consumption. A VMD configuration with 100% thermal efficiency (η) consumes the highest specific thermal energy during desalination due to its poor latent heat recovery from the low-pressure vapor. Thus, the thermal efficiency, i.e., GOR or STEC, largely depends on the extent of heat recovered from the permeate stream or the efficiency of heat integration scheme in the desalination process. It seems that VMD and SGMD configurations retain the potential for improvement in STEC if larger condensation of water vapor in permeate stream can be achieved. 

The important features of each MD configuration are summarized qualitatively in Figure 15. Among various MDs, VMD experiences the highest flux production and thermal efficiency, while AGMD outperforms in terms of latent heat recovery (GOR). The high flux feature of VMD is favorable to achieving high water recovery when multiple VMD modules are parallel connected. In VMD and SGMD, the permeate stream is in the vapor phase; cooling the permeate stream with the feed stream is not sufficient for the total condensation of vapor. As a result, the majority of water vapor remains after exchanging heat to the feed stream, leading to only a partial recovery of latent heat. Permeate streams in VMD and SGMD require additional flashing for liquid water production, which can be an extra energy penalty. In contrast, water vapor condenses internally in DCMD and AGMD (Figure 13). Moreover, the module size of VMD is limited by the high flux and DCMD by the feed salinity, while a large module can be useful for the feasible SGMD and AGMD (Figure 4). Even though SGMD and AGMD have no size constraints, significant flux reduction in a larger module will have an impact on the economic operation of these MDs. The best practice is to develop a desalination process with an optimally designed MD module and subsequent determination of best operation condition and process variables.

## 5. Conclusions

This work presents a comparative study on the performance of four different MD modules for the desalination of hypersaline brine solution, with a special focus on identifying theoretical MD module sizes from the standpoint of mass transfer mechanisms. Calculations performed in this study are supported by rigorous simulation models of MD modules developed in ACM together with an appropriate thermodynamic model for thermophysical and transport property estimation to yield high-fidelity simulation of the MD-based desalination process. Furthermore, flowcharts of the desalination processes are established in Aspen Plus to ensure mass and energy balance closure. Simulated GOR flux in response to permeate conditions for sufficiently large modules can be transferable for the design of MD-based desalination processes on a pilot and industrial scale. The results highlight the importance of maximum extraction of spent heat from the permeate stream for the design of an energy-efficient and high-throughput desalination process with a VMD module.

## Figures and Tables

**Figure 1 membranes-13-00273-f001:**
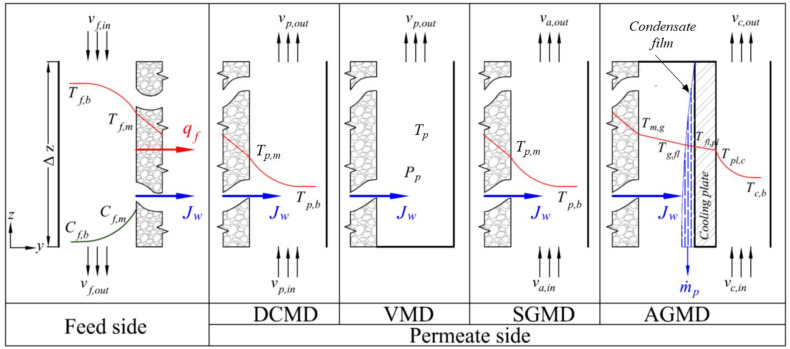
Control volumes of feed and permeate sides for different MD modules. Mass, momentum, and energy balances are made over the control volume of thickness Δz. $J_{w}$ and $q_{f}$ represent mass and heat flux through the membrane, respectively; $T_{f,b}$ and $T_{p,b}$ feed and permeate bulk temperature, respectively; $T_{f,m}$ and $T_{p,m}$ membrane surface temperature on the feed and permeate sides, respectively; $T_{p}$ temperature on the permeate side; $T_{m,g}$, $T_{g,fl}$, $T_{fl,pl}$, $T_{pl,c}$, and $T_{c,b}$ temperature at the membrane-air gap interface, the air gap-condensate film interface, the condensate film-cooling plate interface, and the cooling plate-coolant interface, respectively; $C_{f,b}$ and $C_{f,m}$ concentration in the feed bulk solution and on the feed side of the membrane, respectively; $v_{p,in}$ and $v_{p,out}$ permeate velocity at the inlet and outlet, respectively; $v_{a,in}$ and $v_{a,out}$ air velocity at the inlet and outlet, respectively; $v_{c,in}$ and $v_{c,out}$ coolant velocity at the inlet and outlet, respectively; and $P_{p}$ vacuum on the permeate side.

**Figure 2 membranes-13-00273-f002:**
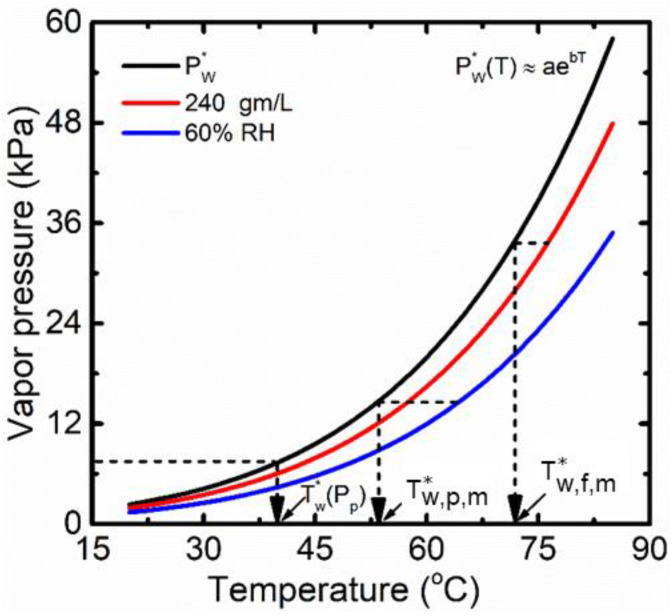
Saturated temperatures of water equivalent to vapor pressure depression, moisture content of air, and vacuum pressure. The values of a and b are 0.988 and 0.0492, respectively, obtained from the exponential curve fitting of vapor pressure vs. temperature from the Antoine equation.

**Figure 3 membranes-13-00273-f003:**
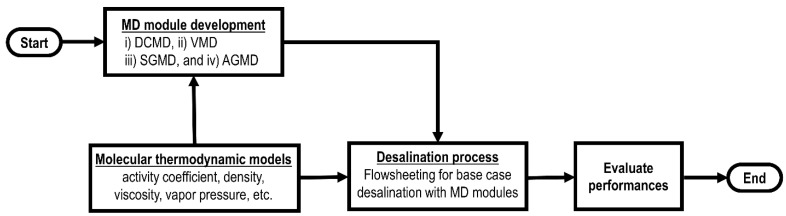
Flowchart for performance calculation of MD-based desalination.

**Figure 4 membranes-13-00273-f004:**
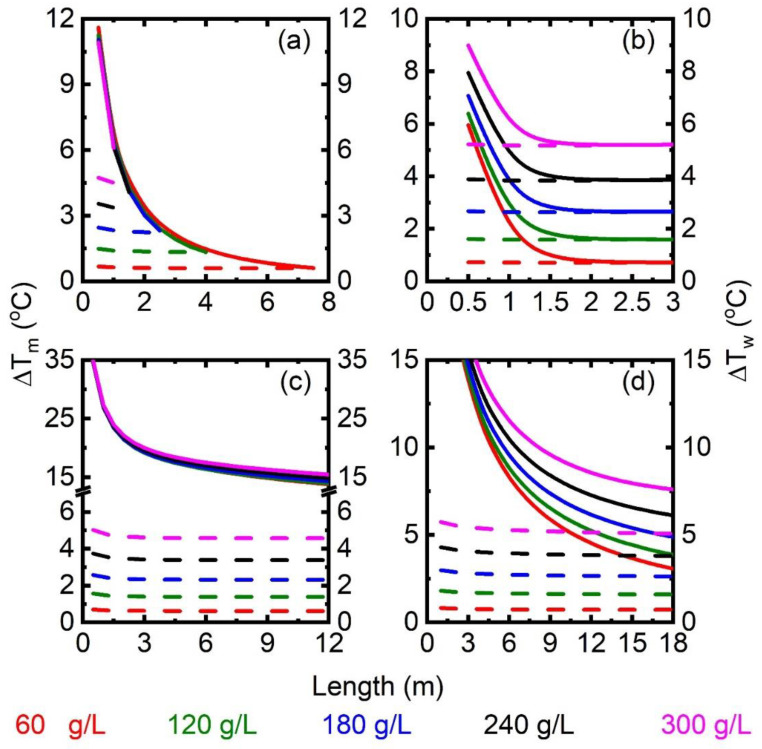
Variation in $\Delta T_{m}$ (solid lines—left y-axis) and $\Delta T_{w}$ (dashed lines—right y-axis) along the length at different feed salinities for (**a**) DCMD, (**b**) VMD, (**c**) SGMD, and (**d**) AGMD.

**Figure 6 membranes-13-00273-f006:**
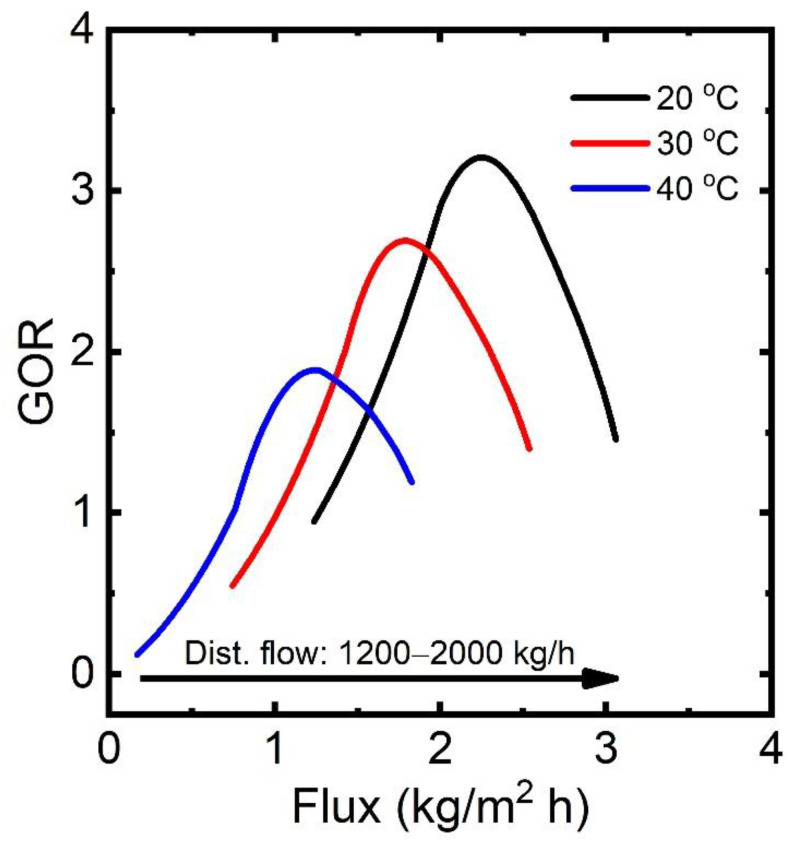
GOR–flux performance for permeate condition in DCMD with varying distillate flow rate of 1200–2000 kg/h and temperature of 20–40 °C. Note that GOR and flux are calculated from the DCMD-based desalination process at each condition.

**Figure 7 membranes-13-00273-f007:**
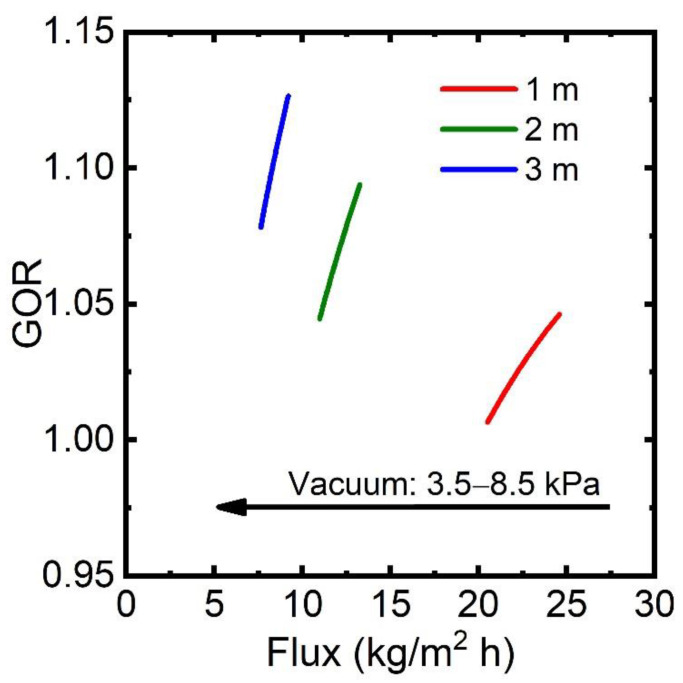
GOR–flux performance for permeate pressure in VMD with varying permeate pressure of 3.5–8.5 kPa and module length of 1–3 m. Note that GOR and flux are calculated from the VMD-based desalination process at each condition.

**Figure 8 membranes-13-00273-f008:**
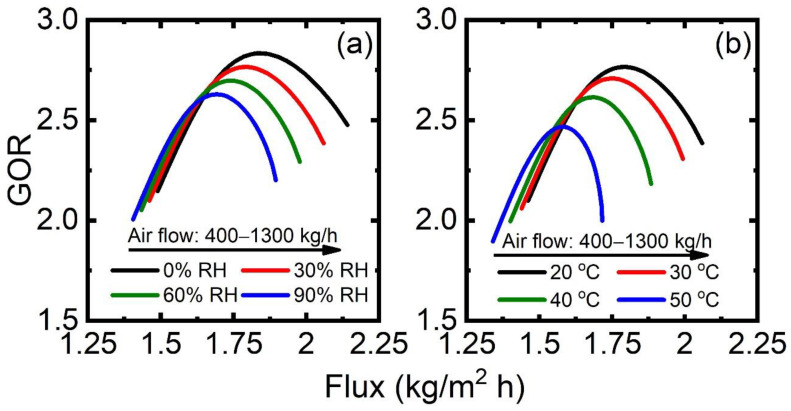
GOR–flux performance for permeate conditions in SGMD with varying (**a**) airflow rate of 400–1300 kg/h and (**b**) RH of 0–90%. Note that GOR and flux are calculated from the SGMD-based desalination process at each condition.

**Figure 9 membranes-13-00273-f009:**
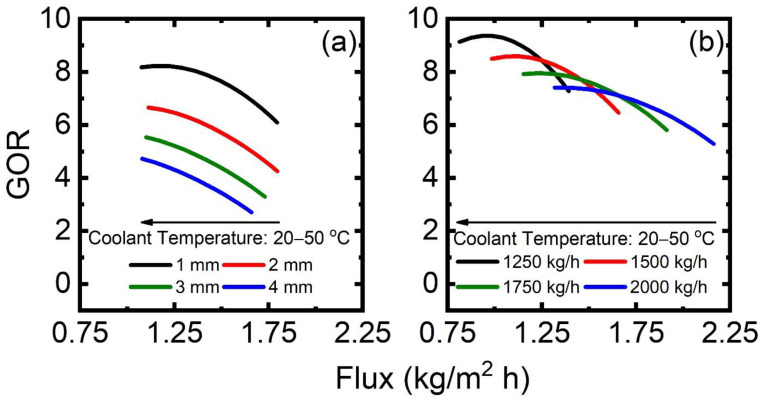
GOR–flux performance for permeate conditions in AGMD with varying coolant temperature of 20–50 °C for (**a**) air gap thickness of 1–4 mm and (**b**) coolant flow rate of 1250–2000 kg/h. Note that GOR and flux are calculated from the AGMD-based desalination process in each condition.

**Figure 10 membranes-13-00273-f010:**
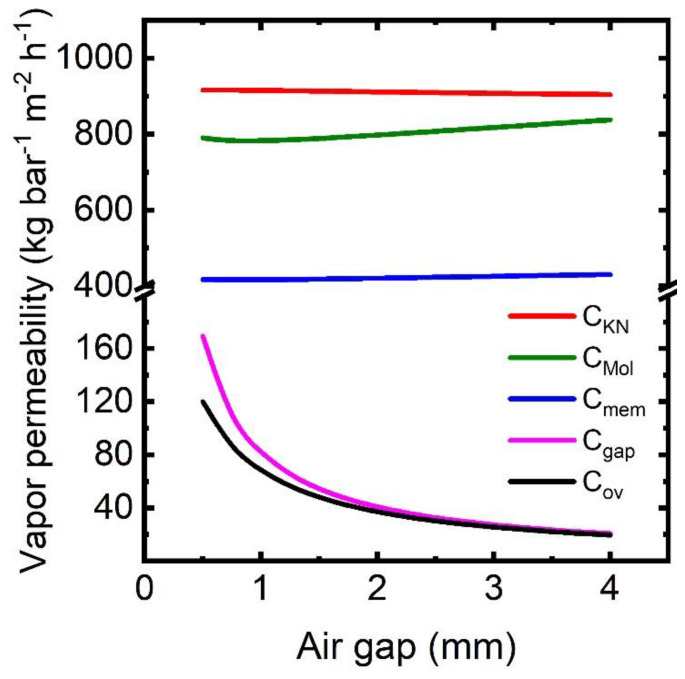
Vapor permeabilities in different mass transport regions of the AGMD process. *C*_KN_, *C*_Mol_, *C*_mem_, *C*_gap_, and *C*_ov_ denote the vapor permeability corresponding to Knudsen diffusion, molecular diffusion, diffusion inside the porous membrane, mass transfer through the air gap, and overall mass transfer through the membrane and air gap.

**Figure 11 membranes-13-00273-f011:**
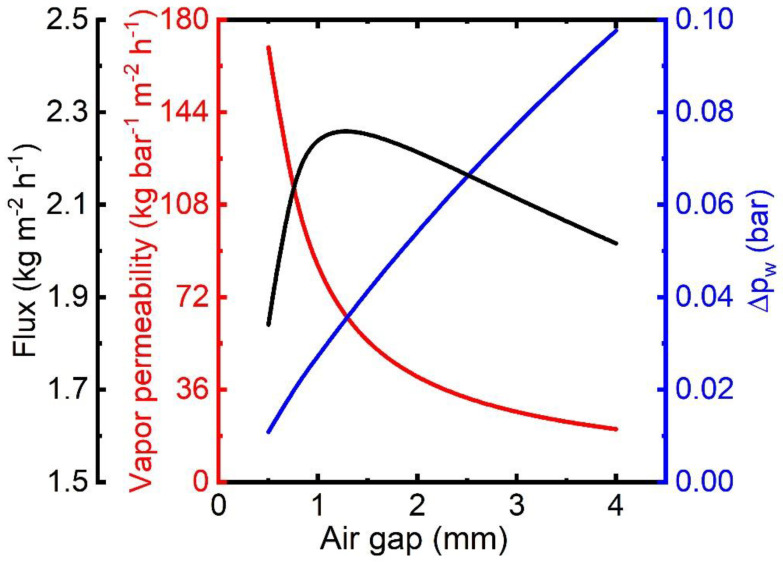
Flux, vapor permeability, and vapor pressure difference profile with varying gap thickness.

**Figure 12 membranes-13-00273-f012:**
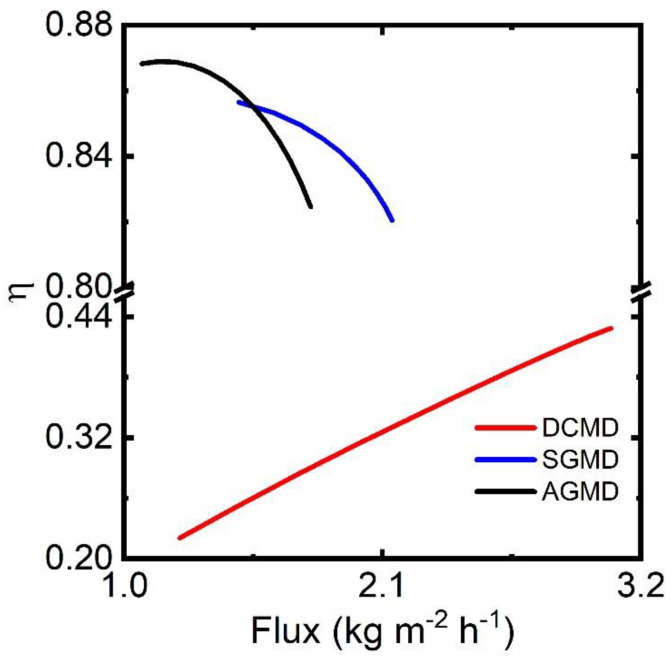
Thermal efficiency (η) of different MD configurations.

**Figure 13 membranes-13-00273-f013:**
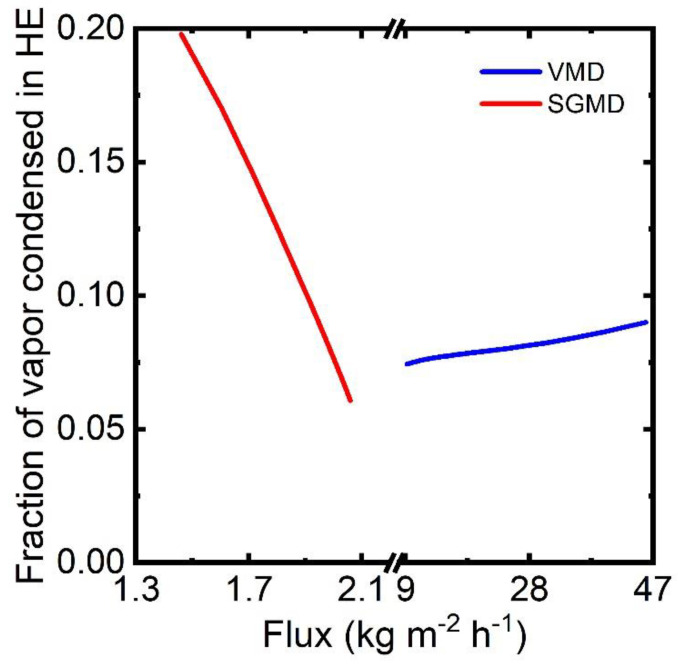
Partial condensation of vapor in VMD and SGMD-based desalination process.

**Figure 14 membranes-13-00273-f014:**
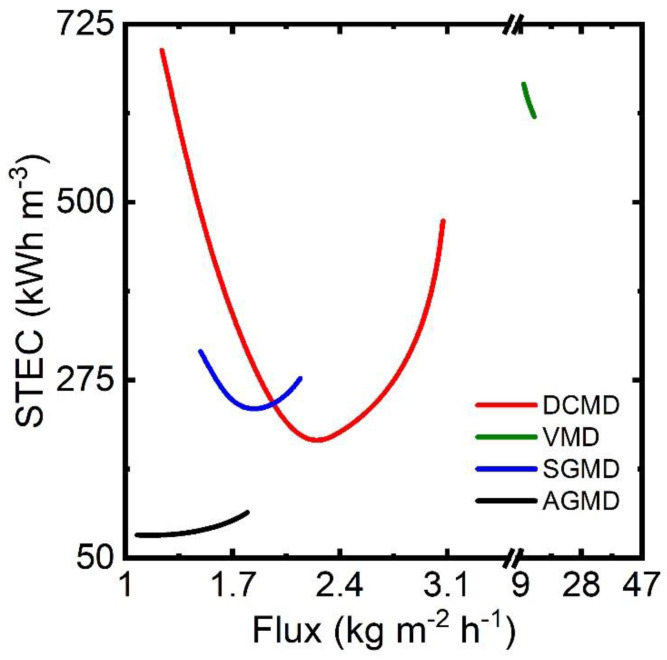
Flux vs. STEC in MD-based desalination.

**Figure 15 membranes-13-00273-f015:**
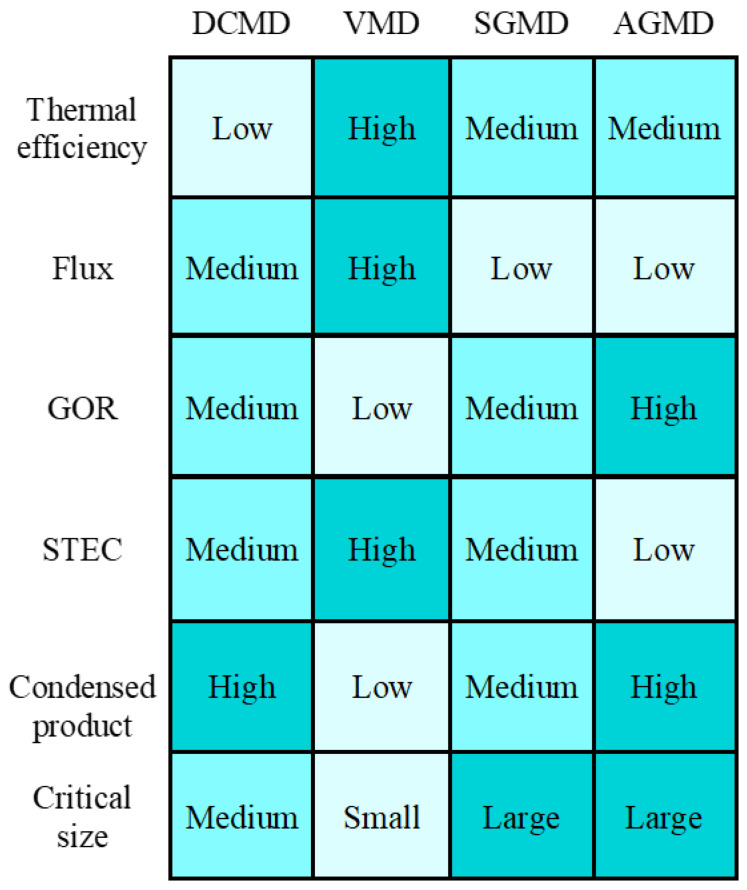
Qualitative summary of various MD configurations versus various key efficiency, operating, and economic parameters.

**Table 1 membranes-13-00273-t001:** Mass transfer mechanism in MDs.

	Diffusion Flux		Viscous Flux
	Knudsen	Molecular	
DCMD	✓	✓	✕
VMD	✓	✕	✓
SGMD	✓	✓	✕
AGMD	✓	✓	✕

**Table 2 membranes-13-00273-t002:** Balance equations in the permeate channel of DCMD and SGMD and coolant channel in AGMD.

	Mass	Momentum	Energy
DCMD	$\frac{dv_{p}}{dz}=-\frac{v_{w}}{h}$	$\frac{dP_{p}}{dz}=\frac{1}{2}f_{p}\frac{\rho_{p}}{d_{h}}v_{p}^{2}$	$\rho_{p}C_{p,p}\frac{d\left(v_{p}T_{p,b}\right)}{dz}=-\frac{q_{p}}{h}$
SGMD	$\frac{d\left(\rho_{a}v_{a}\right)}{dz}=-\frac{J_{w}}{h}$	$\frac{dP_{a}}{dz}=\frac{1}{2}f_{a}\frac{\rho_{a}}{d_{h}}v_{a}^{2}$	$\rho_{a}C_{p,a}\frac{d\left(v_{a}T_{a,b}\right)}{dz}=-\frac{q_{p}}{h}$
AGMD	$\frac{dv_{c}}{dz}=0$	$\frac{dP_{c}}{dz}=\frac{1}{2}f_{c}\frac{\rho_{c}}{d_{h}}v_{c}^{2}$	$\rho_{c}C_{p,c}\frac{d\left(v_{c}T_{c,b}\right)}{dz}=-\frac{q_{c}}{h}$

**Table 3 membranes-13-00273-t003:** Vapor permeabilities in membrane and air gap.

Membrane			Air Gap	System
Knudsen	Molecular	Combined		
$B^{K}=\frac{1}{3}\frac{\varepsilon }{\chi }\frac{d_{p}}{\delta_{m}}\sqrt{\frac{8M_{w}}{\pi RT}}$	$B^{M}=\frac{D_{wa}}{\delta_{m}}\frac{\varepsilon }{\chi }\frac{\bar{P}}{{\left|p_{a}\right|}_{\ln}}\frac{M_{w}}{RT}$	$\frac{1}{B^{m}}=\frac{1}{B^{K}}+\frac{1}{B^{M}}$	$B^{G}=\frac{D_{wa}}{\delta_{g}}\frac{\bar{P}}{{\left|p_{a}\right|}_{\ln}}\frac{M_{w}}{RT}$	$\frac{1}{B^{\;}}=\frac{1}{B^{m}}+\frac{1}{B^{G}}$

**Table 4 membranes-13-00273-t004:** Equivalent saturation temperature ($T_{p}^{*}$) for partial pressure of water in the permeate side.

DCMD	VMD	SGMD	AGMD
$T_{p,m}$	$T_{w}^{*}\left(P_{p}\right)$	$T_{w}^{*}\left(p_{w,p,m}\right)$	$T_{g,fl}{}^{\;}$

**Table 5 membranes-13-00273-t005:** Validation of MD modules.

MD Modules	DCMD	VMD	SGMD	AGMD
Data source	[17]	[16]	[18]	[15]
Number of data	10	4	15	16
Module geometry				
Length (mm)	400	85	85	100
Width (mm)	150	39	39	50
Channel height (mm)	1	2	2	2
Air gap (mm)	–	–	–	5–13
Membrane				
Type	PTFE	PTFE	PTFE	PTFE
Pore size (μm)	0.22	0.20	0.20	0.2
Thickness (μm)	110	30	30	100
Porosity (%)	83	80	80	80
Tortuosity	2.13	2.91	2.91	1.5
Operating condition				
Feed (temperature, salinity, flow rate)	40–60 °C	50–60 °C	40–80 °C	40–80 °C
	10 g/L	0–123 g/L	0 g/L	0–43 g/L
	92–302 kg/h	72–80 kg/h	15–74 kg/h	87–92 kg/h
Permeate (temperature, flow rate, vacuum, and RH)	Distillate	Vacuum	Air 30% RH	Coolant
	93–301 kg/h	6.5 kPa	20 °C	20 °C
	20 °C		0.06–0.35 kg/h	90 kg/h
%ARD	11.63	9.82	12.10	7.44

**Table 6 membranes-13-00273-t006:** Geometry of the modules and operating of MD desalination processes.

MD Modules	DCMD	VMD	SGMD	AGMD
Module geometry				
Length (m)	Depends on MD configurations			
Width (m)	6	6	6	6
Channel height (mm)	2	2	2	2
Air gap (mm)	–	–	–	1
Operating condition				
Feed (feed temperature, flow rate)	85 °C @1634 kg/h			
Feed salinity	60–300 g/L of NaCl			
Permeate	Distillate	Vacuum	Air 30% RH	Coolant
	20 °C	3.5 kPa	20 °C	20 °C
	1634 kg/h		930 kg/h	1634 kg/h

## Data Availability

Data will become available upon request.

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
