# Peer review of "Comparative Energetics of Various Membrane Distillation Configurations and Guidelines for Design and Operation"

_membranes, 2023, doi:10.3390/membranes13030273_

Round 1

Reviewer 1 Report

Dear Authors,

I like very much the presentation and the description of the different forms of MD, as well as the description of the thermal conditions of the different mass and heat transports.  When comparing results with experimental data, reference was made here to various other works. The critical question here is always the scale in which such investigations take place in practice and how far these data then also represent the process in an accurate manner.

In the following I would like to make some question to represented results in this work. In any case, explanations should be given here or changes should be made.

In equation 22, should it not be a temperature difference instead of the condensation temperature Tcond.

A large number of references are missing or incorrect and are displayed as errors!

Line 282: What are these factors a and b, or are they from the Antoine equation?

Figure 10: Can the vapor permeability of the membrane at one bar pressure difference be 400 kg/m2h. This is achieved with water with a much higher density compared to the water vapor only in MF membranes.

Figure 13: For VMD, indicate permeate fluxes from 9 to 47 kg/m2h. These are actually fluxes as we know them with RO. Are these high fluxes, which would have to be achieved in the form of steam through the membrane. Is this possible at all with the gradients given here??

Figure 14: This would also have to be explained for this plot, where STEC values between 9 and 27 L/m2h are also given.  Here it seems to be especially the case that the entire evaporation heat is to be applied for the produced permeate!  In line 538, the VMD ? is specified as 1. Is this not a contradiction with the thermal efficiency?

Thus, it is expressed here that up to 100% (1) of the heat is used for the evaporation of water from the heat supplied by the hot feed solution

Line 535: Here it is shown that ? increases with increasing flux and according to literature [41] ? of 0.94 is reached at permeate flux of 22 kg/m2h.  Could this not also be due to a leak between the feed and permeate side?

In the case of VMD, ? is also given here as 1. In Figure 14, the STEC values are 700 kWh/m3 for fluxes of about 20 L/m2h.

Again, is this not a contradiction and are these flows at all plausible.

Author Response

Please find the response letter attached.

Reviewer 2 Report

This study presents a comparative performance study of single-stage desalination processes with major configurations of membrane distillation (MD) modules. Calculations performed in this study are supported by rigorous simulation models of MD modules developed in ACM together with an appropriate thermodynamic model for thermophysical and transport property estimation. Each model is well described and, genrally speaking, the paper is written well.

I suggest to paper to be accepted only after an implementation of the introduction section. Membrane distillation as desalination technology is well described in the introduction within its advantages and drawbacks in comparison with other filtration techniques. Anyway, the introduction should include some references to the water issue and the need of different purification technologies i.e adsorption, photocatalysis, filtration etc..

Author Response

(The authors gave the same response as above.)

Reviewer 3 Report

Overall Comment:

This manuscript presented by Islam et al. utilized ASPEN to simulate different parameters and performance of four major types of MDs, including DCMD, VMD, SGMD, and AGMD. The computing method using e-NRTL was first validated by comparing the simulated results to literatures. Then many important parameters in different MD configurations were discussed. This work can potentially help to design and predict the performance of MD. Although the authors tried to cover all types of MD, too many equations and unclear definitions impede readers from understanding this work. Moreover, the presentation is not systematically arranged, increasing the difficulty of reading. The article could be more useful if the authors can rearrange the sections and the contents. A major revision is therefore suggested before consideration for publication. Our detailed comments are as follows

1.        There are many citation errors “Error! Reference source not found” that need to be corrected.

2.        [page 6, line141] The authors mentioned Antoine equation is used and mentioned several terms such as “Pi*” and “gi” but not no equation was provided.

3.        There are a few comments for the symbols:

A.        The list of symbols needs reorganization to make it clearer.

B.        The dimensionless unit [-] is used without defined in the list of symbols.

C.        Some of the terms used and explained in the main content is contradictory to the list of symbols. For example, vw in Eq(7) is defined as “permeate velocity”. However, it refers to “water vapor velocity” according the list of symbols.  The author should make them consistent.

D.        Some of the symbols are repeated used for different meaning. For example, f in Eq (9) refers to dimensionless friction factor, but as a subscript it means feed. This is quite confusing and should be prevented.

4.        Since many equations were mentioned, the author should clearly explain the purpose and the meaning of each equation.

5.        In Figure 1, may the authors confirm if the SGMD temperature profile is the same as DCMS? 

6.        [p10, line 269] What does “Tf,m Tp,m” mean? Moreover, the author is suggested to use Figure 1 to explain the temperature drop.

7.        A clearer definition on the MD modules including the length, width, and the channel height should be made. Does the membrane size mean its length and width?

8.        The Figures in Appendix should have a different index from the main content, such as Figure A1 and Figure B1. Figure 2 was not cited.

9.        The authors should be consistent u sing “gm/L” or “g/L” through the manuscript.

10.    Despite the authors have simulate the effects of many parameters on the four types of MD, there has no any specific guidelines for design and operation. This does not coincide with its title. The author is suggested to have a table at the end discussion these.

Author Response

(The authors gave the same response as above.)

Round 2

Reviewer 1 Report

Dear Authors

As I have already explained, I like your work very much, only in the conclusions and derived results should still be some adjustments.

I would recommend to revise the conclusions with the results effectively obtained in this work.

Line 740: This work represents a theoretical study of the performance of four different MD modules. The different module systems were not practically tested with respect to the considerations derived from the simulations.

Line 742 and 747: Identifying appropriate module sizes according to three different lengths and relying on the simulated GOR fluxes as a basis for designing pilot and industrial processes does not seem very convincing to me.

Line 749: Also, I think that the use of the heat of condensation introduced into the permeate to save heat for the evaporation process is obvious and so not a highlight of this work, since this requirement is known and more or less the case for all MD modules.

In the title of this work, guidelines for design and operation are promised. It does not seem quite given to me that this was achieved by the qualitative comparison of MD systems that was made. Also, I cannot find the corresponding guidelines for operation.  Perhaps it would be useful to adapt the title of this work to the actual results obtained.

Line 445: The necessary validation over the works [15, 16, 17 and 18], which were carried out with different objectives (e.g. scaling), I cannot understand unfortunately also completely. Nor is there any evidence that these models have been validated against or with experimental data. I think it should be described more precisely what has been achieved and what can only be described by certain assumptions.

Author Response

(The authors gave the same response as above.)

Reviewer 3 Report

better

Author Response

Thanks.